# Clinical Characteristics and Predictors of All-Cause Mortality in Patients with Hypertensive Urgency at an Emergency Department

**DOI:** 10.3390/jcm10194314

**Published:** 2021-09-22

**Authors:** Jeong-Hun Shin, Byung Sik Kim, Minhyung Lyu, Hyun-Jin Kim, Jun Hyeok Lee, Jin-kyu Park, Young-Hyo Lim, Jinho Shin

**Affiliations:** 1Division of Cardiology, Department of Internal Medicine, Hanyang University Guri Hospital, Hanyang University College of Medicine, Guri-si 11923, Korea; fish3777@hanmail.net (B.S.K.); francel@naver.com (M.L.); titi8th@gmail.com (H.-J.K.); 2Department of Biostatistics, Yonsei University Wonju College of Medicine, Wonju 26426, Korea; ljh0101@yonsei.ac.kr; 3Division of Cardiology, Department of Internal Medicine, Hanyang University Seoul Hospital, Hanyang University College of Medicine, Seoul 04763, Korea; cardiohy@hanyang.ac.kr (J.-k.P.); mdoim@hanyang.ac.kr (Y.-H.L.); jhs2003@hanyang.ac.kr (J.S.)

**Keywords:** hypertensive urgency, emergency department, predictor, mortality

## Abstract

Hypertensive urgency is characterized by an acute increase in blood pressure without acute target organ damage, which is considered to be managed with close outpatient follow-up. However, limited data are available on the prognosis of these cases in emergency departments. We investigated the characteristics and predictors of all-cause mortality in Korean emergency patients with hypertensive urgency. This cross-sectional study included patients aged ≥18 years who visited an emergency tertiary referral center between January 2016 and December 2019 for hypertensive urgency, which was defined as a systolic blood pressure of ≥180 mmHg and a diastolic blood pressure of ≥110 mmHg, or both, without acute target organ damage. The 1 and 3 year all-cause mortality rates were 6.8% and 12.1%, respectively. The incidence of emergency department revisits and readmission after 3 months and 1 year was significantly higher in non-survivors than in survivors. In a multivariate analysis, age ≥ 60 years (hazard ratio (HR), 16.66; 95% CI, 6.20–44.80; *p* < 0.001), male sex (HR, 1.54; 95% CI, 1.22–1.94; *p* < 0.001), history of chronic kidney disease (HR, 2.18; 95% CI, 1.53–3.09; *p* < 0.001), and proteinuria (HR, 1.94; 95% CI, 1.53–2.48; *p* < 0.001) were independent predictors of 3 year all-cause mortality. The all-cause mortality rate of hypertensive urgency remains high despite the increased utilization of antihypertensive medications. Old age, male sex, history of chronic kidney disease, and proteinuria were poor prognostic factors for all-cause mortality in patients with hypertensive urgency.

## 1. Introduction

Physicians in emergency departments (EDs) frequently encounter patients with hypertensive crisis, which is an acute and severe rise in blood pressure (BP) presenting with highly heterogeneous profiles ranging from the absence of symptoms to life-threatening acute target organ damage [1,2,3]. Hypertensive crisis is further classified as either hypertensive emergency or hypertensive urgency based on associated rapid target organ (heart, brain, kidney, and arteries) deterioration. A hypertensive emergency is associated with severe and potentially life-threatening acute target organ damage, thereby requiring hospitalization, preferably in an intensive care unit, for prompt BP control by the intravenous administration of antihypertensive drugs. In contrast, hypertensive urgency is associated with severe BP elevation without acute or impending target organ damage. These patients are treated by the reinstitution or intensification of oral antihypertensive drugs, and there is no indication for immediate BP reduction in an ED or hospitalization [1,2,3,4,5]. Hypertensive emergency and urgency are not completely distinct entities; unrecognized or untreated urgency cases may evolve into an emergency and are still associated with significant morbidity and mortality [6]. However, data concerning short-term and long-term clinical outcomes, as well as clinical characteristics and predictors of all-cause mortality, are scarce in patients with hypertensive urgency. In this study, we aimed to assess the clinical outcomes and predictors of all-cause mortality in patients with hypertensive urgency in an ED.

## 2. Materials and Methods

### 2.1. Study Population

This observational, cross-sectional study was conducted at a single regional emergency medical center: Hanyang University Guri Hospital, Guri-si, Gyeonggi-do, Republic of Korea. We reviewed the medical records of 172,105 patients who visited the ED of this center between January 2016 and December 2019. We included patients diagnosed with hypertensive crisis with an initial triage BP ≥ 180/110 mmHg. Patients with acute trauma or who only needed a medical certificate were excluded; if they visited the ED multiple times, only data from the first visit were included. Patients with hypertensive crisis were further classified based on the presence of acute target organ damage, such as hypertensive encephalopathy, cerebral infarction, intracerebral hemorrhage, retinopathy, acute heart failure, acute coronary syndrome, acute renal failure, and aortic dissection, which was diagnosed based on clinical data and diagnostic tests, such as blood chemistry analysis, eye fundus examination, 12-lead electrocardiography (ECG), chest radiography, echocardiography, computed tomography (CT), and magnetic resonance imaging [7]. Laboratory data included complete blood count, blood biochemical test findings (levels of electrolytes, renal and liver function tests, and blood glucose), routine urinalysis results, and blood levels of C-reactive protein, D-dimer, brain natriuretic peptide, and troponin-I. Proteinuria was defined as a dipstick urinalysis result of ≥1+ [8]. Left ventricular hypertrophy (LVH) was defined based on the Sokolow–Lyon ECG voltage criteria: R-wave in lead V6 ≥ 35 mm or the sum of S-wave in V1 and R-wave in V5 or V6 ≥ 35 mm. Cardiomegaly was defined as a cardiothoracic ratio >0.5 on chest radiography. BP was measured in the ED over the brachial artery using an automated Spot Vital Signs LXi (Welch Allyn, Skaneateles Falls, NY, USA) sphygmomanometer. After excluding patients with hypertensive emergency, 4488 patients with hypertensive urgency were included in the main analysis. Patients were followed up until all-cause death or at the end of the study period (15 March 2021) (Figure 1).

### 2.2. Data Collection and Outcomes

Data were collected using electronic medical records by trained data collectors under the supervision of the principal investigator. The collected data included demographic and clinical characteristics, previous medical history, cardiovascular risk factors and comorbidities, BP, patterns of acute target organ damage, laboratory findings, diagnostic test findings, use of antihypertensive drugs, and events during the hospitalization and follow-up periods (e.g., admission and readmission, discharge, ED revisit, death). The timing and overall incidences of mortality were extracted from the National Health Insurance Service in South Korea.

### 2.3. Statistical Analysis

Categorical variables were tested using a chi-square test or Fisher’s exact test, as appropriate, and presented as numbers and percentages. Continuous variables were compared using a Student’s *t*-test and presented as means with standard deviations (SDs) or medians with interquartile ranges. Multivariate Cox proportional hazard regression analysis was used to identify the predictors of 1 year and 3 year all-cause mortality in patients with hypertensive urgency. Significant (*p* < 0.05) and clinically relevant variables, including baseline characteristics (age, sex, and systolic BP), comorbidities (hypertension, diabetes mellitus, ischemic stroke, hemorrhagic stroke, coronary artery disease, and chronic kidney disease), and components of hypertension-mediated organ damage (HMOD; creatinine level, proteinuria, cardiomegaly on chest radiography, and LVH on ECG), in a univariate analysis were included in a multivariable Cox proportional hazard model, which was constructed using backward elimination, with *p* = 0.1 as the criterion for retention of a variable in the model. Hazard ratios were reported with corresponding 95% confidence intervals. All tests were two-tailed, and statistical significance was set at *p* < 0.05. All analyses were performed using Statistical Analysis Software (version 9.4; SAS Institute, Cary, NC, USA).

## 3. Results

### 3.1. Baseline Characteristics

A total of 4488 patients were enrolled in the study, and follow-up data for up to 5.2 years were analyzed. The median follow-up period was 3.1 years (interquartile range, 2.0–4.1 years). Among these patients, 612 (13.6%) died during the study period. Baseline characteristics of the enrolled patients are shown in Table 1. Their mean (SD) age was 59 (17.3) years, and 51% were women. A total of 2073 (48.4%) patients had hypertension, of which 1149 (55.4%) were taking antihypertensive drugs. Compared with survivors, the non-survivors were older (74.2 vs. 56.6, *p* < 0.001), were more frequently males (54.6% vs. 48.1%, *p* = 0.003), and had a higher incidence of comorbidities such as hypertension, diabetes mellitus, ischemic stroke, hemorrhagic stroke, coronary artery disease, peripheral artery disease, heart failure, and chronic kidney disease. They also showed more abnormal findings associated with HMOD, such as high serum creatinine level, proteinuria, cardiomegaly on chest radiography, and LVH on ECG. The groups did not differ significantly in the frequency of antihypertensive medications taken currently and those administered in the ED.

### 3.2. Outcomes of the Index Visit and during the Follow-Up Period

Overall, 1200 (26.7%) patients were admitted, 2795 (62.3%) patients were discharged, and 490 (10.9%) patients were discharged against medical advice (Table 2). The mortality rate in patients with hypertensive urgency was 3.0%, 6.8%, and 12.1% after 3 months, 1 year, and 3 years, respectively. Non-survivors were significantly more likely to be admitted to the general ward or intensive care units (48.9% vs. 23.3%, *p* < 0.001) and less likely to be discharged from the ED than were survivors (37.7% vs. 66.2%, *p* < 0.001).

The overall rates of ED revisits within 1 month, 3 months, and 1 year were 10.6%, 17.7%, and 30.4%, respectively. The overall rates of readmission within 1 month, 3 months, and 1 year were 7.2%, 9.9%, and 14.7%, respectively. The rates of ED revisits and readmissions in non-survivors were significantly higher than those in survivors.

### 3.3. Predictors of All-Cause Mortality

In the multivariate analysis, age ≥ 60 years (hazard ratio (HR), 16.66; 95% CI, 6.20–44.80; *p* < 0.001), male sex (HR, 1.54; 95% CI, 1.22–1.94; *p* < 0.001), history of chronic kidney disease (HR, 2.18; 95% CI, 1.53–3.09; *p* < 0.001), and proteinuria (HR, 1.94; 95% CI, 1.53–2.48; *p* < 0.001) were independent predictors of 3 year all-cause mortality (Table 3). Additional analysis for 1 year all-cause mortality showed similar results, that age ≥ 60 years (HR, 18.89; 95% CI, 4.66–76.49; *p* < 0.001), male sex (HR, 1.44; 95% CI, 1.07–1.95; *p* < 0.001), and proteinuria (HR, 1.89; 95% CI, 1.53–2.48; *p* < 0.001) were independent predictors (Appendix A).

## 4. Discussion

We investigated the clinical characteristics and predictors of long-term mortality in patients with hypertensive urgency who visited an ED. This study reflects the actual state of hypertensive urgency in EDs in South Korea. The major findings of this study were as follows: (1) all-cause mortality in patients with hypertensive urgency remains high; (2) non-survivors had higher ED revisit and readmission rates and more abnormal findings associated with HMOD, such as a high serum creatinine level, proteinuria, cardiomegaly on chest radiography, and LVH on ECG, than did survivors; and (3) old age, male sex, history of chronic kidney disease, and proteinuria were independent predictors of all-cause mortality in patients with hypertensive urgency.

Hypertensive crisis accounts for an estimated 4.6% of all visits to EDs and is a frequent reason for hospitalization in the United States from 2006 to 2013. Despite improvements in treatment for hypertension in the past decades, the incidence of hypertensive crisis has not declined [6,9,10]. Hypertensive crisis is an important and common event that needs to be well-known among ED staff. Patients undergoing hypertensive emergency should be admitted for close monitoring and, in most cases, treated with intravenous BP-lowering agents to reach the recommended BP target in the designated time frame. In contrast, patients with hypertensive urgency could be treated with oral BP-lowering agents and are usually discharged after a brief observation period [5,6]. The evaluation and therapeutic approaches for hypertensive crisis are well-described by international guidelines [1,2,3,4], but most of them are focused on hypertensive emergencies. There is limited evidence for the evaluation, management, and follow-up strategy related to hypertensive urgencies, which are clinically two to three times more common than hypertensive emergencies. This probably means that clinicians are relatively less interested in hypertensive urgency than in hypertensive emergency, which they perceive as a serious situation.

Previous data on the long-term and short-term outcomes of hypertensive urgency have shown inconsistent results. A prospective study showed that patients with hypertensive urgency admitted to an ED had a 50% higher risk of cardiovascular events during the 5 year follow-up period compared to those without hypertensive urgency, despite similar BP levels during follow-up [11]. Merlo et al. reported that 6% of patients with hypertensive urgency generally experienced a cardiovascular event within 1 year [12]. Guiga et al. documented that the 1 year mortality rate for patients experiencing an episode of hypertensive urgency in an ED at a single center was 8.9% [13]. In contrast to long-term outcomes, increased risks of adverse outcomes during the days to several months after patients were sent home from an outpatient office or ED have not been documented [14,15,16]. Our study showed a much higher mortality rate in patients with hypertensive urgency than expected, reflecting the recent real-world practice data. We previously reported that despite a significantly lower long-term mortality in patients without acute HMOD than in patients with acute HMOD, the values were still very high, with 6.8% and 12.1% after 1 year and 3 years, respectively [7]. In addition, the present study showed that 26.8% of patients with hypertensive urgency were hospitalized during the index visit, and the overall ED revisit and readmission rates were higher than expected: 10.6% and 7.2% within 1 month and 17.7% and 9.9% within 3 months, respectively. Given that many patients with hypertensive urgency presented with mild non-specific symptoms such as dizziness and headaches, medical staff were less likely to be aggressive in performing tests for target organ damage or prescribing treatments than they would have been for patients with specific symptoms, which could lead to an underestimation of target organ damage or discharge of patients who need hospitalization. Although the guidelines recommend the adjustment of antihypertensive medications without further hospitalization for patients with hypertensive urgency, a significant number of patients are hospitalized in actual practice. Despite this, they show a high frequency of ED revisits and readmissions, as well as surprisingly high long-term mortality. Our results suggest that patients with hypertensive urgency require appropriate treatment and close follow-up.

Clinical practice guidelines emphasize the importance of evaluating HMOD and using it as an indicator in risk assessment [1,17]. In this study, HMOD indices such as high serum creatinine level, proteinuria, cardiomegaly on chest radiography, and LVH on ECG were all significantly more marked in non-survivors than in survivors. This may be evidence to highlight the importance of evaluating HMOD, even in highly specified hypertensive patients with acute and severe rise in BP.

In this study, the prognostic factors for 3 year all-cause mortality were old age, male sex, history of chronic kidney disease, and proteinuria. Individuals with chronic kidney disease are at high risk of cardiovascular disease, progression to end-stage renal disease, and all-cause mortality [18]. Chronic kidney disease is an established cardiovascular risk factor in patients with hypertension, and hypertensive patients with chronic kidney disease are considered to be at high cardiovascular risk. In view of the high prevalence of chronic kidney disease in individuals with hypertension, clinical guidelines recommend screening for chronic kidney disease in hypertensive individuals and more intensive intervention in patients with chronic kidney disease to prevent adverse outcomes [1,2,3]. Our results suggest that more intensive treatment and follow-up strategies are needed for patients with hypertensive urgency with chronic kidney disease. Proteinuria is a strong marker for renal injury because it can be detected before any perceptible decline in eGFR. Although the use of the urine albumin-to-creatinine ratio is recommended in evaluating renal damage in all patients with hypertension, the urine dipstick test also has high sensitivity and specificity in screening for proteinuria [8,19]. Oh et al. also reported that proteinuria, defined as ≥1+ on a dipstick test, was a powerful independent risk factor for all-cause death in patients with hypertension [20], which was corroborated by our findings that proteinuria is an independent prognostic factor for long-term mortality in patients with hypertensive urgency. Further studies are needed to clarify the prognostic differences according to the degree of proteinuria, the feasibility of the screening test, and the improvement of outcomes after proteinuria treatment in patients with hypertensive urgency.

Taken together, our findings showed that even if there are no symptoms indicative of target organ damage, an in-depth subclinical HMOD evaluation is required in patients with hypertensive urgency visiting an ED. A routine metabolic panel for assessing renal function and electrolyte levels, complete blood count testing, urinalysis for identifying proteinuria, and ECG and assessment of troponin levels for ruling out asymptomatic myocardial injuries are needed. Parallel to careful HMOD assessment, improving adherence and persistence are pivotal in reducing the risk of complications and recurrent hospitalization for hypertensive urgency. Future research regarding the optimal screening, risk stratification, treatment strategy, and follow-up interval, as related to the short-term and long-term clinical outcomes, is needed.

This study had several limitations. Firstly, this was a retrospective study. Although the study was based on reliable registry data and electronic medical charts, compared with the accuracy and completeness of data used in prospective studies, retrospective data on baseline characteristics of the study populations were insufficient. In addition, we could not obtain data on socioeconomic status or the awareness, treatment, and control rate of hypertension, reason for hospitalization, and control rate of hypertension after their ED visits. Secondly, the study included data from a single center, which may not be representative of the entire population. Additionally, data regarding outpatient follow-up and ED revisit and readmission rates could have been underestimated. Thirdly, we could not identify cardiovascular events and cardiovascular mortality in this study because the National Health Insurance Service data did not provide information on the cause of death. However, data regarding all-cause mortality and date of death were highly accurate because they were obtained from the National Health Insurance Service, which covers the entire population of Korea. Fourthly, we defined proteinuria as ≥1+ on dipstick testing and investigated it as a binary variable. Therefore, analysis of the clinical significance of microalbuminuria and differences in mortality according to the degree of proteinuria was not possible. Finally, diagnostic tests for target organ damage were not performed in all patients, and it is likely that more tests were performed in relatively high-risk patients than in low-risk patients, so the possibility of selection bias cannot be excluded.

## 5. Conclusions

The all-cause mortality of patients with hypertensive urgency remains high despite the increased utilization of antihypertensive drugs. Old age, male sex, history of chronic kidney disease, and proteinuria were independent predictors of 3 year all-cause mortality in patients with hypertensive urgency.

## Figures and Tables

**Figure 1 jcm-10-04314-f001:**
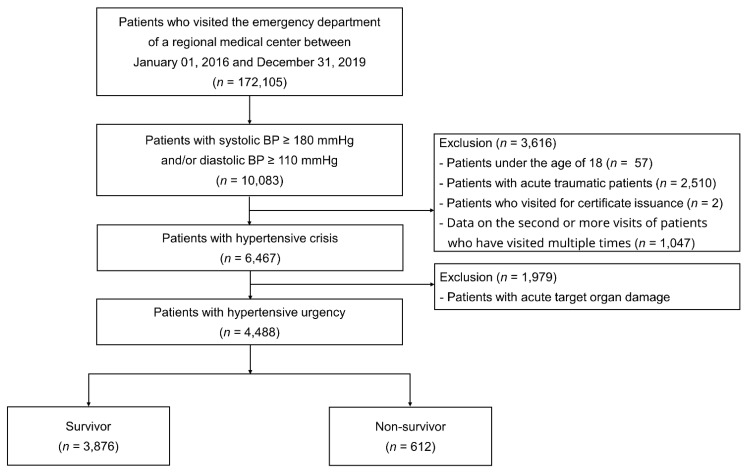
Study flowchart.

**Table 1 jcm-10-04314-t001:** Baseline characteristics.

	All Patients (*n* = 4488)	Survivors (*n* = 3876)	Non-Survivors (*n* = 612)	*p*-Value
Age, years, mean (SD)	59.0 (17.3)	56.6 (16.7)	74.2 (12.9)	<0.001
Males, *n* (%)	2200 (49.0)	1866 (48.1)	334 (54.6)	0.003
Medical history, *n* (%)				
Hypertension	2073 (48.4)	1688 (45.8)	385 (64.5)	<0.001
Diabetes mellitus	985 (23.3)	748 (20.6)	237 (40.3)	<0.001
Dyslipidemia	351 (8.4)	323 (9.0)	28 (4.8)	<0.001
Ischemic stroke	267 (6.4)	176 (4.9)	91 (15.6)	<0.001
Hemorrhagic stroke	98 (2.4)	71 (2.0)	27 (4.7)	<0.001
Coronary artery disease	277 (6.7)	219 (6.1)	58 (10.0)	<0.001
Peripheral artery disease	39 (0.9)	28 (0.8)	11 (1.9)	0.010
Heart failure	72 (1.7)	43 (1.2)	29 (5.0)	<0.001
Chronic kidney disease	253 (6.1)	151 (4.2)	102 (17.5)	<0.001
Social history, *n* (%)				
Cigarette smoking	545 (22.8)	479 (24.3)	66 (15.8)	<0.001
Alcohol consumption	822 (33.5)	745 (36.8)	77 (18.2)	<0.001
Triage vitals, mean (SD)				
SBP, mmHg	186 (20.1)	186 (19.9)	191 (20.9)	<0.001
DBP, mmHg	108 (16.1)	109 (15.7)	102 (17.3)	<0.001
Laboratory tests done, *n* (%)	3579 (79.7)	3061 (79.0)	518 (84.6)	0.001
Serum creatinine, mg/dL, mean (SD)	1.12 (1.5)	1.04 (1.3)	1.62 (2.0)	<0.001
eGFR, mL/min/1.73 m^2^, mean (SD)	84.6 (29.0)	88.0 (27.0)	66.0 (32.2)	<0.001
Urinary analysis done, *n* (%)	2194 (48.9)	1862 (48.0)	332 (54.2)	0.004
Proteinuria, *n* (%)	626 (28.5)	463 (24.9)	163 (49.1)	<0.001
Chest radiography done, *n* (%)	3392 (75.6)	2885 (74.4)	507 (82.8)	<0.001
Cardiomegaly, *n* (%)	449 (12.9)	340 (11.4)	109 (21.1)	<0.001
Congestion/fluid overload, *n* (%)	4 (0.1)	3 (0.1)	1 (0.2)	0.565
ECG done, *n* (%)	3094 (68.9)	2585 (66.7)	509 (83.2)	<0.001
LVH, *n* (%)	300 (9.7)	232 (9.0)	68 (13.4)	0.002
Myocardial ischemia, *n* (%)	117 (3.8)	85 (3.3)	32 (6.3)	0.001
Atrial fibrillation, *n* (%)	102 (3.3)	67 (2.6)	35 (6.9)	<0.001
Brain imaging done, *n* (%)	1246 (27.8)	1089 (28.1)	157 (25.7)	0.210
Abnormal findings, *n* (%)	85 (5.4)	71 (5.2)	14 (6.8)	0.340
Chest and abdomen CT done, *n* (%)	437 (9.7)	320 (8.3)	117 (19.1)	<0.001
Echocardiography done, *n* (%)	20 (0.4)	17 (0.4)	3 (0.5)	0.859
Fundoscopy done, *n* (%)	71 (1.6)	65 (1.7)	6 (1.0)	0.199
Abnormal findings, *n* (%)	7 (1.7)	6 (1.7)	1 (1.9)	0.918
Current antihypertensive medication, *n* (%)	1149 (55.4)	921 (54.5)	228 (59.2)	0.212
ED antihypertensive medication, *n* (%)	836 (18.6)	710 (18.3)	126 (20.6)	0.180
IV nicardipine	381 (8.5)	316 (8.2)	65 (10.6)	0.042
IV labetalol	13 (0.3)	11 (0.3)	2 (0.3)	0.854
IV esmolol	5 (0.1)	4 (0.1)	1 (0.2)	0.678
IV nitroglycerin	124 (2.8)	91 (2.4)	33 (5.4)	<0.001
Oral calcium-channel blocker *	395 (8.8)	355 (9.2)	40 (6.5)	0.033
Oral beta-blocker ^†^	66 (1.5)	57 (1.5)	9 (1.5)	1
Oral renin-angiotensin system inhibitor ^↑^	19 (0.4)	16 (0.4)	3 (0.5)	0.784
Oral nitroglycerin	108 (2.4)	98 (2.5)	10 (1.6)	0.180

Data are presented as *n* (%) or mean (SD), as appropriate. SD, standard deviation; SBP, systolic blood pressure; DBP, diastolic blood pressure; eGFR, estimated glomerular filtration rate; ECG, electrocardiography; LVH, left ventricular hypertrophy; CT, computed tomography; ED, emergency department; IV, intravenous. +Proteinuria was defined as a dipstick urinalysis result ≥ 1+. * Calcium antagonists included amlodipine and nifedipine. ^†^ Beta blockers included carvedilol, nebivolol, propranolol, atenolol, and bisoprolol. ^↑^ Renin-angiotensin system inhibitors included perindopril, candesartan, losartan, and fimasartan.

**Table 2 jcm-10-04314-t002:** Outcomes of the index visit to the emergency department of a regional medical center and during the follow-up period.

	All Patients (*n* = 4488)	Survivor (*n* = 3876)	Non-Survivor (*n* = 612)	*p*-Value
Hospital outcomes, *n* (%)				
Admission	1200 (26.7)	901 (23.2)	299 (48.9)	<0.001
Discharge	2795 (62.3)	2564 (66.2)	231 (37.7)	<0.001
Discharge against medical advice	490 (10.9)	411 (10.6)	79 (12.9)	0.089
ED revisit within the time period, *n* (%)				
1 month	349 (10.6)	279 (10.0)	70 (14.4)	0.004
3 months	582 (17.7)	448 (16.0)	134 (27.5)	<0.001
1 year	999 (30.4)	774 (27.6)	225 (46.2)	<0.001
Readmission within the time period, *n* (%)				
1 month	237 (7.2)	196 (7.0)	41 (8.4)	0.272
3 months	327 (9.9)	261 (9.3)	66 (13.5)	0.004
1 year	485 (14.7)	376 (13.4)	109 (22.3)	<0.001
Mortality within the time period, *n* (%)				
1 month	79 (1.8)			
3 months	134 (3.0)			
1 year	303 (6.8)			
3 years	542 (12.1)			

Data are presented as *n* (%). ED, emergency department.

**Table 3 jcm-10-04314-t003:** Predictors for 3 year all-cause mortality.

Variables	Univariate		Multivariate	
	HR (95% CI)	*p*-Value	Adjusted HR (95% CI)	*p*-Value
Age (vs. <40 years)				
40 to 59 years	4.05 (1.86–8.80)	<0.001	2.38 (0.83–6.82)	0.106
≥60 years	22.08 (10.47–46.57)	<0.001	16.66 (6.20–44.80)	<0.001
Male sex	1.33 (1.13–1.58)	<0.001	1.54 (1.22–1.94)	<0.001
SBP (per 1 mmHg)	1.01 (1.01–1.02)	<0.001		
History of hypertension	2.03 (1.70–2.43)	<0.001		
History of diabetes mellitus	2.24 (1.88–2.67)	<0.001		
History of ischemic stroke	3.22 (2.55–4.07)	<0.001		
History of hemorrhagic stroke	1.95 (1.27–2.99)	0.002		
History of coronary artery disease	1.58 (1.18–2.11)	0.002		
History of chronic kidney disease	4.08 (3.26–5.11)	<0.001	2.18 (1.53–3.09)	<0.001
Creatinine (per 1 mg/dL)	1.15 (1.11–1.20)	<0.001		
Proteinuria	2.72 (2.17–3.42)	<0.001	1.94 (1.53–2.48)	<0.001
Cardiomegaly on chest radiography	1.94 (1.55–2.43)	<0.001		
LVH on ECG	1.57 (1.20–2.05)	<0.001		

HR, hazard ratio; CI, confidence interval; SBP, systolic blood pressure; LVH, left ventricular hypertrophy; ECG, electrocardiography.

## Data Availability

All data of this study can be requested from the corresponding author (cardio.hyapex@gmail.com).

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
