# Peer review of "Clinical Characteristics and Predictors of All-Cause Mortality in Patients with Hypertensive Urgency at an Emergency Department"

_jcm, 2021, doi:10.3390/jcm10194314_

Round 1
Reviewer 1 Report
This study by Shin et al. is a single center retrospective study that assesses the clinical predictors of all-cause mortality in patients who present with hypertensive urgency to the ED. I applaud the authors' efforts to address this important question. The authors identify age, male sex, history of chronic kidney disease, and proteinuria as independent predictors of 3-year all-cause mortality.
I have several suggestions that will improve the study:
- Please describe the proportion of patients within the survivor and non-survivor groups that had well-controlled blood pressures after their ED visits and what was the average follow-up systolic and diastolic pressures.
- Non-survivors had higher rates of hospitalization after their ED visits. What was the reason for the hospitalizations? Was it due to hypertensive urgency or another reason? Please also describe the reasons for presentation to the ED. Were most cases incidental discoveries of hypertensive urgency?
- In Figure 1, 1047 patients were excluded due to visiting the ED multiple times. However, in the methods, it notes that if patients visited the ED multiple times, only data from the first visit was analyzed. Please clarify this discrepancy.
Author Response
Because, our response to the reviewer’s comments has the tables and figure,
so we upload an attachment.

Reviewer 2 Report
Hypertensive crisis is a frequent reason for consultation in the ER in either forms: urgency or emergency.
Hypertensive urgency is more frequent than hypertensive emergency and is usually considered to have a good prognosis. Authors interestingly show that patients presenting with a hypertensive urgency have also an unfavorable prognosis with a 12% mortality at 3 years.
The strong points of this work are the duration of the follow up, the large number of patients included and the robust main outcome: occurrence pf death as specified by the national health insurance system.
Some limitations exist and have been pointed out by the authors in the discussion chapter. Indeed, beyond the retrospective design, given the data collection strategy (ER files), several information are lacking such as the detailed antihypertensive drugs (pharmaceutical classes) and whether hypetension was controlled or un controlled before and after ER consultation. There is obviously a lot of heterogeneity in this population as it was included on the basis of BP value and regardless of the reason for consultation (only trauma were discarded). Furthermore, the study prespecified main outcome was all-cause death, but the cause of death remained unknown. It is impossible to know whether this high mortality was related to hypertension complications or not.
Despite these shortcomings, the paper remains interesting and brings a precious information on the mortality of patients suffering from hypertensive urgency, which is clearly a marker of worse prognosis in hypertensive patients.
Author Response
Point 1: Some limitations exist and have been pointed out by the authors in the discussion chapter. Indeed, beyond the retrospective design, given the data collection strategy (ER files), several information are lacking such as the detailed antihypertensive drugs (pharmaceutical classes) and whether hypetension was controlled or un controlled before and after ER consultation. There is obviously a lot of heterogeneity in this population as it was included on the basis of BP value and regardless of the reason for consultation (only trauma were discarded). Furthermore, the study prespecified main outcome was all-cause death, but the cause of death remained unknown. It is impossible to know whether this high mortality was related to hypertension complications or not. 

Response 1: We would like to thank the reviewer for this valuable comment. We totally agree with your opinion that contents mentioned by the reviewer are limitations of our study, and if data were present, the contents would have been more delicate. We think that further well-designed prospective studies are needed to include data on the rate of blood pressure control, the reasons for blood pressure rise, and the cause of death. Thank you again for suggesting a good research topic.
And, we added some of more limitation in the discussion section (page 9, lines 255, 256) as follows:
“In addition, we could not obtain data on socioeconomic status or the awareness, treatment, and control rate of hypertension, reason of hospitalization, and control rate of hypertension after their ED visits."